# Fermentation Kinetics of Gluten-Free Breads: The Effect of Carob Fraction and Water Content

**DOI:** 10.3390/foods12091809

**Published:** 2023-04-27

**Authors:** Kleopatra Tsatsaragkou, Ioanna Mandala, Nikolaos G. Stoforos

**Affiliations:** Laboratory of Food Process Engineering, Department of Food Science and Human Nutrition, Agricultural University of Athens, 11855 Athens, Greece; kleotsat@gmail.com (K.T.); stoforos@aua.gr (N.G.S.)

**Keywords:** protein, fiber, yeast, model, bakery products

## Abstract

In this study, gluten-free doughs with rice flour, substituted by 15% fractions of different carob seed flours, were prepared by varying their water content. The coarse carob fraction A (median particle size of flour, D_50_: 258.55 μm) was rich in fibers, fraction B (D_50_: 174.73 μm) was rich in protein, C (D_50_: 126.37 μm) was rich in germ protein, and fraction D (D_50_: 80.36 μm) was a mix, reconstituted from the other fractions and pulverized using a jet mill. Τhe experimental data of the dough’s volume over time were fitted to the Gompertz model for each carob fraction and water content. The calculated parameters of the model were the maximum relative volume expansion ratio (*a*), the maximum specific volume growth rate (*μ*), and the time lag of the leavening process (*t_lag_*). Gompertz’s equation adequately described the individual experimental curves. In the next step, a composite model was applied for each carob fraction where the parameters *a* and *t_lag_* were expressed as quadratic functions of water content levels (*W*), while *μ* was linearly dependent on *W*. Each carob fraction presented an optimum water content level for which dough height was maximized and time lag was minimized. Optimized dough volume could be predicted by the composite model; it was shifted to lower values as finer carob flour was used. In respect to baked products, softer breads were produced using finer carob flour and porosity values were higher at optimum water content levels. The investigated fermentation kinetics’ models provide significant information about the role of water and carob flour on gluten-free dough development and bread volume expansion.

## 1. Introduction

The basic steps of the breadmaking procedure are mixing, kneading, dough fermentation, and baking. During mixing, the ingredients are transformed into a viscoelastic matrix during kneading, and air bubbles are dispersed and act as the initial nuclei for the gas bubbles, which will develop during the subsequent stages. During fermentation, yeast metabolizes flour sugars into carbon dioxide gas which diffuses toward the air nuclei during kneading, causing their growth. The viscoelastic matrix developed during mixing is able to retain the gas produced during the fermentation process, yielding an aerated crumb bread structure [1]. Proteins contribute to gas cells’ successful expansion because they control their growth rate. Gas cells coalesce at the latest stages of fermentation, and proteins prevent an extensive creation of large cells that lead to a gas loss during baking [2]. For wheat breads, the gluten matrix is capable of maximum gas retention [3], while in gluten-free breads, the absence of structure-forming gluten proteins results in dough with poor resistance to mechanical stress and gas retention during a long fermentation time. As a result, fermentation time is reduced in gluten-free doughs, causing gluten-free bread to have generally lower palatability and a shorter shelf life compared to traditional bread [4]. On the other hand, starch granules are embedded in the protein matrix, and, according to Hayman et al., are considered inclusions [5]. The particle size of the starch determines the coalescence of the gas cells. Their walls thin during baking and coalesce with adjacent gas cells, when their thickness becomes smaller than the diameter of the adjacent starch granules [5]. This suggests that large-size granules, such as those of the potato, result in low cell wall stability, and the final bread structure is characterized as an open crumb grain [5]. Recently, different treatments are suggested to improve starch’s impact on dough fermentation. The ultrasonication of frozen dough at low ultrasonic power densities results in starch granules’ structural changes that promote starch–yeast interactions and lead to increased fermentation rates. The HHP treatment of different starches causes fermentation time increases in gluten-free doughs, suggesting a fine structure of the final product [6,7].

Dietary fibers or proteins are largely used in the preparation of gluten-free breads as structure-forming agents are necessary to rebuild the protein network [8]. Carob bean flour contains increased amounts of dietary fibers and proteins, and it is considered a safe substitute for gluten-free products. Carob germ proteins have been shown to have functional properties similar to wheat gluten, since they can form a true dough instead of a batter [9]. Commonly, gluten-free baked products are made from batters and not from dough-based systems, resulting in limited overall processability.

The addition of dietary fibers in baked products mainly leads to specific volume reductions and alterations in sensorial characteristics. Flour particle size is considered to affect flour behavior in breadmaking, since it alters the rheological properties of the dough. More specifically, sorghum-based gluten-free bread characteristics were significantly affected by flour particle size and composition [10]. According to de la Hera et al. [11], gluten-free maize bread made with coarser particle size flour exhibited increased volume and reduced firmness compared to breads made with the finer flour fractions. The particle size of chestnut flour also affected the rheology of gluten-free dough [12].

The addition of bran at high concentrations results in lower gas cell sizes in every stage of the dough process and its superfine grinding results in decreased CO_2_ production during fermentation and in a low final bread volume. Bran particles also hinder the bubble evolution, particularly their coalescence in the latter stages of fermentation [13]. It is also interesting that flours rich in fiber (breadfruit flour) reduce the specific volume of the gluten-free bread because they influence the gas cells’ thickness. Additionally, the leavening agent, yeast or baking powder, has an impact on the color and sensorial attributes of the breads [14].

The rheological properties of gluten-free dough also affect dough gas retention and tolerance to fermentation. Controlling and understanding the rheological properties of dough and monitoring the dough fermentation process will boost the production of high-quality gluten-free bakery products. An assessment of the expansion ratio of dough during fermentation can be monitored using devices such as a rheofermentometer or by monitoring the change in dough volume vs. the time using digital imaging techniques [1].

Chevallier et al. [15] compared three different methods of measuring the expansion of bread dough during fermentation. Two of the methods, the horizontal method and the free expansion method, are based on analyzing digital images taken during fermentation, while the third method, the vertical method, is based on monitoring a rise in the height of dough placed in a graduated cylinder. The authors reported that only small differences were observed between the different methods. The vertical method is the most commonly used method and can be used for gluten-free doughs which resemble batters and cannot be subjected to free expansion during fermentation [16].

Fermentation kinetics, influenced by composition and water content, are decisive for the final dough volume and gluten-free bread’s crumb texture. Kinetics change because fibers and/or proteins differ at ratio and size in different flour fractions [16]. It is of special interest to understand fermentation kinetics when a size reduction in fibers, initiated by further grinding, changes their role in the dough. Furthermore, fermentation kinetics modeling provides us with the opportunity to use simple mathematical equations to understand and predict dough and bread expansion, where several factors are taken into account.

The objective of the present study was to investigate the effect of the particle size of carob flour on the fermentative ability of gluten-free doughs with varying the water content. The variation in dough volume for the different carob flour particle sizes and water contents was modeled according to the modified Gompertz equation proposed by [1]. Gluten-free breads with carob flour of different particle sizes and water contents were produced and characterized in terms of physical properties (crumb firmness, crumb porosity, and crumb grain characteristics). An investigation of the carob flour particle size effect on gluten-free dough fermentation and functionality in gluten-free breads can provide information that supports the production of higher-quality gluten-free baked goods.

## 2. Materials and Methods

The materials used in this study were rice flour (moisture content—13.10 g/100 g, protein—7.39 g/100 g, dietary fiber—0.5 g/100 g, lipid—0.39 g/100 g, and ash—0.8 g/100 g, composition measured by [16]) (Kaplanidis mill group S.A., Serres, Greece), moist yeast (L’hirondelle, Group. Lesaffre, Marcq-en-Barœul, France), table sugar, salt (iodized sea salt, Kalas, Athens, Greece), shortening (Vitam, Unilever S.A, Athens, Greece), egg white powder (Laffort S.A., Bordeaux, France), whey protein concentrate (Nutrilac^®^ DR-7015, Arla Foods Ingredients, Viby, Denmark) with 65 g/100 g set as the minimum protein content, emulsifier (DATEM: diacetyl-tartaric esters of mono- and diglycerides, Danisco, Copenhagen, Denmark), locust bean gum (LBG) (Sigma-Aldrich Chemie GmbH, Taufkirchen, Germany), enzymes of alpha-amylase with additional transglutaminase and hemicullolase activity (VERON CLX AB Enzymes, Darmstadt, Germany), tap water, and carob flour. Carob seeds were received from local Cypriot producers and milled in a laboratory attrition mill. Carob flour was sieved and fractions above 500 μm were discarded. The fractions obtained were named as fraction A (315–500 μm), fraction B (250–315 μm), and fraction C (125–250 μm). Carob flour below 500 μm, which was the whole carob seed flour, apart from the large pieces (above 500 μm) of the seed’s peel, was passed through a Jet-O-Mizer Milling (Model 0101S, Fluid Energy Processing and Equipment Company, Telford, PA, USA) with an air pressure at 8 bar, in order to obtain ultra-fine flour powder (fraction D).

### 2.1. Carob Flour Characterisation

Moisture content was determined by Method 925.10 of the American Association of Cereal Chemists (AOAC) [17], while the protein and dietary fiber content was determined as described in [16]. Flour particle size distribution analysis has been performed using a Malvern Mastersizer 2000, equipped with a Scirocco dry powder accessory (Malvern Instruments, Worcestershire, UK). This set-up allowed for the direct exposure of dry particles to the laser beamline. The optical constant used for the particle size measurements was a refractive index of 1.53. Particle size distribution results were cut off between 0.02 and 2000 μm.

### 2.2. Dough Expansion Measurements

Duplicate 10 mL samples of dough from each recipe were placed into a lubricated (vegetable oil) calibrated glass cylinder mold, allowing the dough to expand only vertically, and left for incubation (fermentation) for 50 min at 35 °C and 85% RH. Once the dough was placed inside the cylinder mold, no additional manipulation was made. During the fermentation process, volume expansion data were obtained using the calibrated marks on the mold, with a precision of ±0.5 mL. Experiments were carried out in duplicate.

### 2.3. Breadmaking and Bread Characterization

The basic ingredient for the production of gluten-free breads was rice flour. Carob flour substituted the rice flour by 15% (carob–rice flour ratio: 15/85 based on weight). The basic dough recipe and the breadmaking procedure were described in our previous work [16]. The amount of water added to the dough varied between 100% and 160%, based on the total flour weight for fraction A and between 100% and 150% for the other carob fractions (B, C, and D). All baking tests were performed in two replicates. After baking, the loaves were removed from the pans, and then cooled at room temperature for 1 h before being evaluated.

The firmness of bread crumb was estimated with an Instron Universal Machine (Universal Testing Machine, Model 1100, Instron, Norwood, MA, USA), equipped with a 50 N load cell. A slice of 2.5 cm (in thickness) from the center of the loaf was compressed to 40% of its initial height with a 4 cm diameter probe and a speed of 101 mm/s. The force reading (N), measured at 40% of compression, expressed the resistance of the crumb to the penetrating probe and represented the crumb firmness. Measurements were taken in duplicate.

For crumb porosity measurements, cube samples (1.5 cm in size) were taken from the geometric center of the crumb from all breads. The volume of solids (VS, m^3^) was measured with gas pycnometer (Stereopycnometer SPY-3, Quantachrome, Syosset, NY, USA) using helium as the displacement fluid for volume measurements. Measurements were taken in triplicate. The crumb cellular structure, known as the crumb grain or visual texture, was evaluated. Measurements were conducted on bread slices with a thickness of 1 cm, scanned using a flatbed scanner (HP scanjet 4370, Hewlett–Packard, Palo Alto, CA, USA) and analyzed using image analysis software (Image-Pro Plus 7, Media Cybernetics, Rockville, MD, USA). Values determined were the cell density (cells/cm^2^), average pore size (area), and standard deviation (SD) of the average pore size (uniformity). The standard deviation of the average pore size was considered to be indicative of pore uniformity. The lower SD in the pore area meant that the average pore size was more alike, indicating higher uniformity. The bread specific volume (cm^3^/g) was determined by a solids’ displacement method using glass beads, 2 mm in diameter. A mean of four loaves for each fraction was used.

### 2.4. Mathematical Modeling

The fermentation dough kinetics were investigated by calculating the relative change in the dough volume, (*V*(*t*) – *V*_0_)/*V*_0_, as a function of time;·*V*(*t*) represents the dough volume at time *t* and *V*_0_ represents the initial dough volume. Dough expands due to carbon dioxide production caused by yeast and its migration toward the air nuclei formed in the dough during the mixing step. Consequently, the curves of dough volume expansion have the same shape as the growth curves of yeasts since the carbon dioxide production rate is related to the metabolism of yeast. The growth curves are generally of sigmoid shape and one of the most common equations for their description was initially developed by Gompertz to describe human mortality rates and subsequently to describe the growth of microorganisms in predictive microbiology [1]. Romano et al. [1] modified Gompertz’s equation in terms of quantities that could be related to the fermentation process:(1)$$y(t)=\alpha \exp(-\exp(\frac{\mu e}{\alpha }(t_{lag}-t)+1))$$
with
(2)$$y\left(t\right)=\ln\left(\frac{V(t)-V_{0}}{V_{0}}\right)$$
where *α* represents the maximum relative volume expansion ratio of the loaf; *μ* represents the maximum specific volume growth rate, *t_lag_* represents the time lag of the leavening process, and *e* represents the Euler’s number. According to the model, each fermentation curve exhibits three steps: the induction, the growth, and the stationary phase.

Dough expansion data of each recipe were fitted using both the nonlinear (NLIN) procedure of Systat 10.2 (Systat Software, Inc., Richmond, CA, USA) abd the modified Gompertz model to estimate parameters *α*, *μ*, and *t_lag_*. In a next step, parameters *α*, *μ*, and *t_lag_* were considered functions of the water content of breads. The measurements were statistically analyzed using the Statgraphics Statistical Graphics System, Centurion XV.II (Statgraphics, Rockville, MD, USA). The sample means differed significantly for a *p*-value less than 0.05, according to Fisher’s LSD analysis.

## 3. Results and Discussion

### 3.1. Flour Characterization

Table 1 shows the composition and physical properties of the studied carob flour fractions. The volume’s median diameter d_50_ is the value of the particle size which divides the population exactly into two equal halves, i.e., 50% of the distribution above this value and 50% below. The mean particle size of flour decreased with the decreasing mesh size of sieves. Significant differences in the compositions of carob fractions were observed. With regards to the A, B, and C fractions, the protein content was higher in the finer fractions, unlike dietary fiber content, which was higher in coarser fractions. Based on their weight, carob seed constituents are seed coat (23–33%), endosperm (42–56%), and embryo (20–25%) [18]. The decreasing trend of dietary fiber content from coarser to finer flour fractions (A–C fractions) can be explained by the fact that the seed constituents exhibit different friability [19]. Seed coat and endosperm are harder than embryos [20] and are therefore difficult to be milled in fine granulometry.

A reduction in the carob flour particle size from 258 μm to 126 μm caused a reduction in the water-holding capacity, while an increase in the water-holding capacity of the finer fraction D was observed. The particle size reduction in dietary fiber has been associated with a lower ability to retain water [21], due to the damaging of the fiber matrix and the collapse of the pores during grinding [22]. However, it has also been speculated that in the absence of a matrix structure, a reduction in the particle size might expose a large surface area, and simultaneously expose more polar groups with water binding sites to the surrounding water, increasing the hydration properties of fibers [23,24]. The results of the hydration properties of different types of fibers in the literature are contradictory. According to Zhu et al. [25], the ultrafine grinding of wheat bran dietary fiber decreased its hydration properties, on the contrary to others [26,27], who found that the micronization of carrot insoluble dietary fiber and ginger powder was associated with higher hydration properties. Thus, the effect of particle size on the hydration properties of dietary fibers cannot be generalized and must be assessed for each type of fiber [28].

The hydration properties of dietary fibers in flours are not only determined by fiber size, as its chemical composition also plays an essential role. Caroubin, the protein isolated from the carob bean embryo, is considered to absorb ≈3 g water/g of protein (at 25 °C) [29], while locust bean gum is considered to bind 10 g water/g of gum [30]. Knowledge of the hydration properties is important for the food industry because they influence ingredients’ functionality, product yield, and shelf stability, which are particularly important in the case of baked goods, where water takes part in starch gelatinization phenomena, protein unfolding, and yeast activation during mixing and baking [27,31].

### 3.2. Dough Expansion Modeling

Parameters *a*, *μ*, and *t_lag_* were calculated by fitting the Gompertz model, i.e., Equation (1), to the experimental relative change in the dough volume data, *y*(*t*), vs. time data (Equation (2)) for each carob fraction and water content) (Table 2). Gompertz’s equation adequately described the individual experimental curves, giving R^2^ values between 0.92 and 0.99, with an exception for Fraction D at the intermediate water contents (of 120% and 130%). In a next step, parameters *a* and *t_lag_* were expressed as functions of water content in an effort to derive a composite model. Based on the general trends observed for the kinetic values for each carob fraction and water content, parameters *a* and *t_lag_* were considered to be quadratic functions of the water content level (*W*), while *μ* was considered to be linearly dependent on *W* (Table 3). Incorporating these equations for *t_lag_*, *μ*, and *a* into Equation (1), a composite model, describing the evolution of relative change in dough volume as a function of the water content level, was developed. For each carob fraction, the estimated parameters of the quadratic and linear models are presented in Table 3. The overall performance of the composite model was satisfactory (R^2^ between 0.90 and 0.97, for the four carob fractions, as shown in Table 3).

The *t_lag_* parameter followed a quadratic relationship with water, identifying a minimum lag phase duration point for a specific water content. The variation in the lag phase with water addition can be explained by the role of yeast in the fermentation process. More specifically, during the lag phase, the volume variation relative to the onset of yeast activity derives from both the time taken for the yeast to start fermenting the sugars and the time taken for the CO_2_ to diffuse through the dough matrix to the air nuclei [32]. As a result, an increase in the water content may lead to an increased growth of yeast cells, reducing at the same time the lag phase duration. Water content is crucial in the breadmaking procedure, since dough must be adequately viscous to retain carbon dioxide produced during fermentation and also sufficiently elastic to allow the bubbles to expand and agglomerate with other bubbles [33]. Therefore, an increase in the water levels in the gluten-free dough above the point that corresponds to a minimum lag phase decreases its viscous character, leading to an escape of the produced CO_2_ from yeast and a delayed volume increment (higher values of *t_lag_*). The role of water in retaining a balance between the viscous and elastic properties of gluten-free dough containing carob flour was well documented in a previous study [34].

Parameter *μ* represents the specific volume growth rate which, according to the Gompertz model, is associated with the growth phase where the yeast growth rate increases, monotonically reaching an inflection point at the beginning of the stationary phase. The parameter *μ* is regarded as a linear function of water content, since the increase in water is considered to increase the growth rate of yeast cells, and thus the bread volume growth rate. According to Marechal et al. [35], a decrease in the water activity of the cultivation medium of yeast (*Saccharomyces cerevisiae*) lead to a decrease in the cell viability and cell volume.

The effect of water content was also depicted at the parameter *a*, which represents the maximum relative volume expansion ratio. Higher *a* values were reported for carob fraction B, followed by fractions A and C. The dough prepared with fraction D exhibited *a* values similar to those for doughs made with fraction A. Crockett et al. [36] reported that the addition of soy proteins altered functionality levels in a gluten-free dough containing hydroxypropyl methyl cellulose (HPMC) due to the competition for water and weakened HPMC interactions with the starch matrix reducing foam stability. This may explain the low *a* values for bread made with fraction C, since the presence of carob germ restricted the functionality of locust bean gum from the carob endosperm.

Variations in the dough volume with time for each carob fraction can be seen in Figure 1. Between carob fractions A, B, and C, which were obtained with conventional milling, fractions A and B with a coarser particle size exhibited higher volume developments than fraction C with a finer particle size. Concerning carob fraction D, although it is the finest particle flour, dough development values similar to those obtained by carob fraction A can be observed. This fact can be attributed to the different chemical compositions of fractions. Fraction C was richer in protein from the carob germ, which is considered to restrict dough development. It has been reported [37] that the addition of carob germ protein at a 1.5% level in gluten-free bread batter resulted in the lowest batter height after 3 h of development compared to chickpea, pea, and soya batters. Herein, breads containing carob fraction C presented a 4% increase in the protein content. Furthermore, breads made from carob flour fractions A, B, and D exhibited increased dough height since those fractions were rich in locust bean gum from the endosperm of the seed. The addition of locust bean gum and emulsifier (DATEM) in rice breads improved the volume of breads by allowing the entrapment of air bubbles in dough [38]. Following the addition of water in the recipes, an optimum water content level was observed, namely 150%, 140%, 120%, and 130% levels for carob fractions A, B, C, and D, respectively, for which dough height during fermentation was maximized. The optimum water content level was lower in finer carob flour fractions C and D compared to the coarser ones. This fact can be attributed either to the lower water-holding capacity of fraction C or to the possible altering of the fiber matrix because of the jet milling application for fraction D. It can be speculated that the influence of mixing during breadmaking may led to a reverse trend between the water-holding capacity of carob flour fraction D and the optimum water absorption capacity of the gluten-free dough. According to the literature [21], processing practices such as stirring could alter the physical structure of fibers and thus result in large changes in flour hydration properties.

### 3.3. Bread Characterization

The firmness values of gluten-free breads prepared with the different carob fractions can be seen in Figure 2. The statistical significance levels of two independent factors, carob fraction and water content, which both influence firmness, were tested using a two-way ANOVA test. Water content significantly affected crumb firmness (*p* = 0.0000). An increment of water content led to a decrease in firmness values for all breads made with every carob fraction. The decrease in firmness values for breads made with carob fraction A can be observed above the 140% water content level, on the contrary to all other carob fractions where differences in crumb firmness exhibited an initial sharp decrease at low water content levels and reach a plateau above about 130% water. Water content is a significant parameter for gluten-free breadmaking. It has been reported [4] that increasing water addition in gluten-free breads enriched with diary proteins led to end-products with a much softer crust and crumb texture, which can be attributed to the reduction in the starch retrogradation rate as a result of the presence of extra water.

The granulometry of carob fractions (d_50_) also exhibited a significant effect (*p* = 0.0000) on crumb firmness. A decrease in carob flour particle size led to breads with decreased firmness values. This effect can be attributed to the different dietary composition of the fractions. Coarse fractions have a different composition compared to fine ones. Breads made with the coarse fraction A contained higher amounts of dietary fibers, mainly consisting of polysaccharides (locust bean gum), compared to breads made with fraction C, which are richer in proteins. Gums were commonly used in gluten-free breadmaking to impart texture and appearance properties [8]. According to the literature [39,40], the use of different binding agents (xanthan, guar gum, locust bean gum, and tragacanth gum) in gluten-free bread formulations resulted in a highly significant increase in the loaf volume and the loosening of the crumb structure with the optimum gum concentration varying at levels of 1–3%. Optimum levels for locust bean gum were 2–4% for wheat breads with an increased loaf height [41]. A possible explanation of the increased firmness values for breads made with carob fraction A could be the fact that they contain almost 9.8% carob fiber (based on fraction A composition) (calculated on the dietary fiber content of fraction A), i.e., a much higher concentration from the upper limits of the reported optimum addition of gums. On the other hand, breads made with carob fraction C contained approximately 4% carob germ (based on fraction C composition) (calculated on the protein content of fraction C). Carob germ protein addition in gluten-free formulations at low concentrations lowered the bread firmness values [9]. Except for the chemical composition of the different fractions, the particle size itself affected the texture of the produced breads. Although differences between fractions B and D in protein and dietary fiber content levels were not significant, breads prepared with fraction D presented significantly lower values of crumb firmness. As a rule, the incorporation of fibers in bread reduces loaf volume and increases firmness, but the use of different milling fiber fractions can improve bread quality [42]. The incorporation of smaller-sized dietary fibers from sugarcane in wheat bread gave a more tender and elastic crumb than coarser fibers [20].

Porosity is a significant attribute of high-quality bread and increased porosity values are desirable. Values of porosity for the gluten-free breads that were studied are illustrated in Figure 2. Two-way ANOVA indicated the statistically significant influence of water content (*p* = 0.0001) and d_50_ of the fractions (*p* = 0.0000) on porosity values. An increase in water content seemed to increase porosity values up to a water level, beyond which porosity remained unaffected or reduced. A positive correlation (at *p* < 0.001) between the crumb porosity and water content of gluten-free bread containing amaranth flour has been reported [43]. This trend is more obvious for breads with fraction D, in which porosity evolution follows the dough development curve. Reducing the carob flour particle size led to reduced porosity values at a given water content. Similar observations were reported for bran addition in wheat bread since the bran-reduced particle size decreased the bread volume [44]. The results of dough development measurements are closely related to breads’ physical properties. The water content values that optimize porosity are compatible with the water content values that maximize dough volume development, specifically the *a* values. More specifically, water content levels of 150%, 140%, 120%, and 130% were considered adequate for producing gluten-free breads with carob fractions A, B, C, and D, respectively, possessing adequate dough development, crumb expansion, and low firmness.

The crumb grain characteristics of the produced breads are presented in Figure 3a–c. It can be observed that both water content and carob particle size significantly affected the cell density (*p* = 0.0000 and *p* = 0.0001, respectively), average pore size (*p* = 0.0001 and *p* = 0.0017, respectively), and uniformity of the produced breads (*p* = 0.0001 and *p* = 0.0045, respectively). An increment in the water addition led to a significant decrease in the number of pores (cell density) accompanied by an increase in the average pore size. It is well documented that a high water content can promote the formation of large bubbles in the bread crumb, also resulting in a reduction in crumb firmness [43], a finding in accordance with the results of this study (Figure 3a). At low water content levels (100–110%), cell density exhibited high and average pore size low values, leading to a dense crumb structure with increased firmness.

The uniformity index could be related to the standard deviation of the average pore size; therefore, higher uniformity index values indicate higher variations in the pore diameter size, and thus a broad pore distribution, but also a more open structure. Uniformity decreased following an increase in the water content, leading to an uneven bread crumb, due to the presence of large pores (Figure 3b,c). Breads made with carob fractions A and C were characterized by the most even crumb (low uniformity index values).

Cell size greatly influences the mouthfeel of a bread product; uniformly sized cells yield a soft and elastic bread texture, i.e., properties that are usually welcomed by consumers [45]. Thus, a high number of medium-sized average pores that are uniformly distributed is desirable. Based on the abovementioned analysis, breads with water content levels of 150%, 140%, 120%, and 130% for carob fractions A, B, C, and D, respectively, which were considered to possess good-quality characteristics in terms of dough development and physical properties, were also characterized by improved crumb grain characteristics. A visualization of crumb characteristics of prepared breads with different water content is depicted in Figure 4. When the water content was low, loaf expansion was restricted, and crumb structure was dense, characterized by an increased number of smaller pores. On the other hand, a higher amount of water led to a more open structure with larger pores.

Specific volumes of breads for each carob fraction at the optimum water levels were described in previous studies [46] and can be seen in Table 4. Breads prepared with fractions A and D exhibited the highest specific volumes compared to breads prepared with fractions B and C. Based on the variations in dough volume for each carob fraction with elapsing time (Figure 1), it can be seen that the water content which led to the highest dough volume also produced end-products with acceptable specific volumes, similar to values reported for gluten-free breads by other researchers [47]. Judging from the crumb characteristics of breads, the presence of a higher number of large cells has been correlated to low bread volume, while a high number of medium cells has been found to produce loaves with a high specific volume [4]. In this context, breads prepared with fractions A and D exhibited a lower average pore size and higher cell density compared to breads prepared from fractions B and C, thus explaining the higher specific volumes of breads prepared with fractions A and D.

## 4. Conclusions

Gluten-free doughs and breads were produced with different carob flour fractions and varying the water content levels. A composite model based on Gompertz’s equation was proposed to predict the dough volume change in each fraction in respect to fermentation time and water content. Breads with water content levels of 150%, 140%, 120%, and 130% for carob fractions A, B, C, and D, respectively, were considered to possess a high dough development and good bread physical properties. Considering the above values, each fraction exhibited different fermentation optimum conditions. Fraction A was rich in fibers and dough showed a steady volume increase over time, whereas produced breads were firm with an even crumb grain. Fraction B was rich in protein and presented a high dough volume, a soft crumb, and an open crumb grain. Fraction C was rich in germ proteins and was characterized with high *t_lag_* during fermentation and a restricted volume expansion at high fermentation times. The crumb was soft and the crumb grain was even. Fraction D was a fine fraction of the whole seed composition. Fermentation kinetics strongly depended on water content presenting a maximum dough increase at intermediate water values. The bread crumb was soft and the crumb grain was uneven at high water content levels. Complementary experiments will allow for the replacement of regression equations used to describe the kinetic parameters as functions of water content levels with mechanistically derived equations.

## Figures and Tables

**Figure 1 foods-12-01809-f001:**
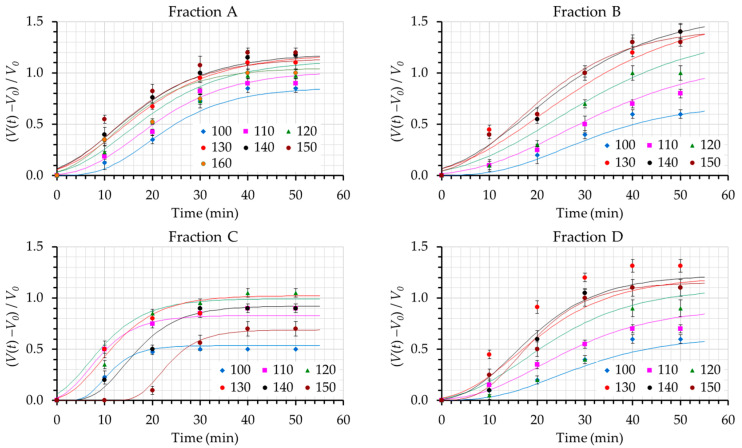
Evolution of the relative change in dough volume, (*V*(*t*) − *V*_0_)/*V*_0_, as a function of the water content levels (110, 120, 130 140, 150, and 160) for the different carob fractions examined (A, B, C, and D). Points refer to experimental data while lines represent values that are fitted through the composite model. Each color refers to a particular water content. Error bars refer to one standard deviation.

**Figure 2 foods-12-01809-f002:**
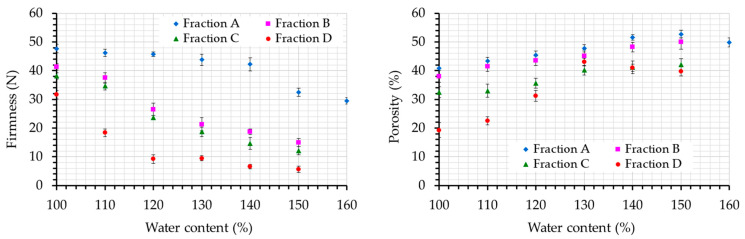
The effect of water content levels on bread firmness (**left**) and bread porosity (**right**) for the different carob fractions examined (A, B, C, and D). Error bars refer to one standard deviation.

**Figure 3 foods-12-01809-f003:**
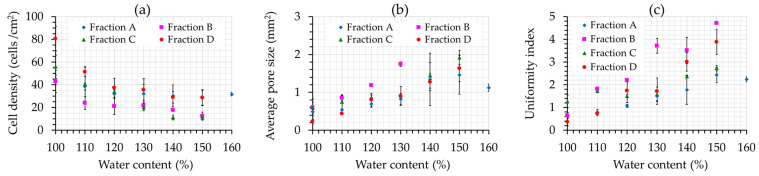
The effect of water content on the cell density (**a**), average pore size (**b**), and uniformity (**c**) of the produced breads for the different carob fractions examined (A, B, C, and D). Error bars refer to one standard deviation.

**Figure 4 foods-12-01809-f004:**
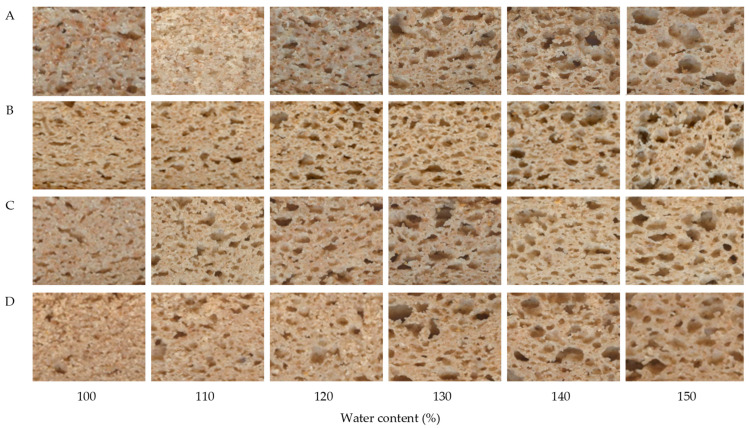
Bread images depicting the effect of water content levels on crumb structure for the different carob fractions (A, B, C, and D) examined.

**Table 1 foods-12-01809-t001:** The composition and physical properties of different carob flour fractions.

Fraction	d_50_ (μm)	Moisture (%)	Protein Content (%)	Dietary Fiber Content (%)	Water-Holding Capacity %
A	258.55 ^a^ (0.68)	9.74 ^c^ (0.03)	15.54 ^a^ (1.51)	65.61 ^c^ (2.34)	702.06 ^b^ (12.53)
B	174.73 ^b^ (0.45)	9.19 ^a^ (0.03)	22.69 ^b^ (1.38)	51.80 ^b^ (2.67)	561.07 ^a^ (25.05)
C	126.37 ^c^ (2.10)	9.31 ^a^ (0.01)	26.64 ^c^ (0.84)	43.46 ^a^ (1.45)	577.70 ^a^ (17.61)
D	80.36 ^d^ (6.38)	9.52 ^b^ (0.18)	21.51 ^b^ (1.17)	53.25 ^b^ (1.64)	800.72 ^c^ (36.01)

^a,b,c,d^ Entries with same letters within each column indicate no significant differences between reported values at a significance level of 0.05. Values in parentheses indicate the standard deviations of the mean values.

**Table 2 foods-12-01809-t002:** Kinetic parameters describing the relative change in dough volume vs. time data, based on Gompertz’s equation, Equation (1), for each carob fraction and water content level studied.

Fraction	% Water Content	*t_lag_*	*μ*	*α*	SSE	R^2^
A	100	8.9650	0.0368	0.9123	0.0191	0.97
	110	6.5236	0.0372	0.9644	0.0146	0.98
	120	3.6838	0.0320	1.0585	0.0417	0.95
	130	2.2920	0.0408	1.1510	0.0061	0.99
	140	1.8023	0.0445	1.2013	0.0106	0.99
	150	0.5964	0.0523	1.1930	0.0758	0.93
	160	1.7090	0.0300	1.1105	0.0721	0.94
B	100	6.8360	0.0197	0.7283	0.0197	0.94
	110	8.9983	0.0245	0.9593	0.0132	0.98
	120	11.9520	0.0413	1.1078	0.0506	0.95
	130	0.9153	0.0337	1.6677	0.0653	0.95
	140	2.6393	0.0368	1.6938	0.0298	0.98
	150	2.1268	0.0383	1.4823	0.0313	0.98
C	100	7.3878	0.0425	0.5010	0.0043	0.98
	110	1.1255	0.0543	0.8848	0.0122	0.98
	120	4.1818	0.0608	1.0423	0.2450	0.95
	130	1.7323	0.0600	0.8790	0.0511	0.92
	140	6.3150	0.0423	0.9483	0.0241	0.97
	150	18.6648	0.0588	0.7118	0.0068	0.99
D	100	8.3965	0.0198	0.7130	0.0279	0.92
	110	4.9568	0.0248	0.7623	0.0166	0.96
	120	15.7343	0.0350	1.0960	0.2450	0.70
	130	2.8528	0.0607	1.2755	0.3036	0.79
	140	9.6047	0.0650	1.1657	0.0357	0.97
	150	5.9925	0.0430	1.1893	0.0307	0.97

**Table 3 foods-12-01809-t003:** Regression equation parameters describing the effect of % water content (*W*) on the kinetic parameters of Gompertz’s equation (*t_lag_*, *μ*, and *a*) for each carob fraction studied.

*t_lag_* = *l_o_* + *l*_1_·*W* + *l*_2_·*W*^2^	Fraction	*l* _o_	*l* _1_	*l* _2_
	A	101.25	−1.4174	0.004983
	B	125.31	−1.7842	0.006466
	C	237.13	−4.0012	0.016885
	D	143.28	−2.1649	0.008418
*μ_max_* = *μ_o_* + *μ*_1_·*W*	Fraction	*μ* _o_	*μ* _1_	
	A	0.01903	0.0001409	
	B	−0.02824	0.0004693	
	C	0.06263	−0.0001048	
	D	−0.05144	0.0006939	
*α* = *α_o_* + *α*_1_·*W* + *α*_2_·*W*^2^	Fraction	*α* _o_	*α* _1_	*α* _2_
	A	−3.531	0.06938	−0.0002546
	B	−11.835	0.19923	−0.0007381
	C	−9.670	0.16803	−0.0006598
	D	−7.003	0.12026	−0.0004393
Fraction	A	B	C	D
SSE	0.233	0.210	0.150	0.659
R^2^	0.97	0.97	0.97	0.90

**Table 4 foods-12-01809-t004:** Specific volumes of breads at the optimum water levels for each carob flour fraction.

Fraction	Water Content (%)	Specific Volume (cm^3^/g)
A	150	2.22 ^b^ (0.07)
B	140	1.70 ^a^ (0.07)
C	120	1.61 ^a^ (0.12)
D	130	2.21 ^b^ (0.13)

^a,b^ Entries with the same letters indicate no significant differences between reported values at a significance level of 0.05. Values in parentheses indicate the standard deviations of the mean values.

## Data Availability

Data can be provided upon request.

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
