# Peer review of "Fermentation Kinetics of Gluten-Free Breads: The Effect of Carob Fraction and Water Content"

_foods, 2023, doi:10.3390/foods12091809_

Round 1

Reviewer 1 Report

The presented material seems interesting and requires partial revision.

The authors need to make minor corrections to improve the article.

Comments are provided below.

1. Form a generalized scientific goal of research

2. For all raw materials used, indicate the city of origin in addition to the country.

3. Authors should submit photographs of received bread samples

4. The authors should give a more detailed description of the storage of bread samples and after what time they were evaluated

Reviewer 2 Report

 The article is well written and discussed and the topic developed is of interest for the journal.

The results are interesting and I have some suggestions to improve the conclusions obtained

A very important parameter in the quality of gluten-free breads is the volume reached by the dough pieces after baking. In this reviewer's experience, an increase in dough volume during the proofing process is not always associated with higher volume of loaves due to changes that occur during the baking of the batter/dough.

For this reason, I suggest that the authors determine the specific volume of the breads and relate it to the water content for each carob flour fraction and to the parameters of the mathematical model applied during fermentation.

It is also suggested to incorporate images of the breads and the structure of the crumb of the breads.

Minor comments:

Line 192. “Referring to A, B, and fraction, protein content..”. It seems some word is missing

Line 234. “The overall performance of the composite model was satisfactory (R2 values between 0.90 and 0.97, Table 3)”. It is not clear for this reviewer how the authors obtained the overall performance. I suggest to add for each regression the R2.

Reviewer 3 Report

The manuscript "Fermentation kinetics of gluten-free breads. Effect of carob fraction and water content" can contribute in enhancing gluten-free breads and other products production, and simultaneously enriched these products with various nutrients.

Organization of manuscript and experiment is good. Abstract is concise and clear for readers to understand the aim of the experiment as well as obtained results. Introduction provide sufficient background. Methods used in this research are adequately described. Obtained results are clearly presented and explained, in some parts can be improved, which I will listed below. References can be improved with more recent date references.

 Suggestions for manuscript corrections:

Line 11-12; line 116-117; line 408-409 – this fraction D should be described more clearly

Line 192 “Reffering to A, B, and fraction, protein…” – this sentence is not clear enough, what fraction (except A and B)?

Line 192-193 Interpretation of the results does not match with results presented in Table 1. Fraction D is the finest fraction and does not have the higher protein content

Line 194-195 Interpretation of the results does not match with results presented in Table 1. The decreasing trend of dietary fiber content is from A to C fractions. In fraction D (the finest flour) content of dietary fiber is higher

Line  233 "...are presented in Table 3."
